# Parasympathetic and Sympathetic Monitoring Identifies Earliest Signs of Autonomic Neuropathy

Nicholas L. DePace [1,2], Luis Santos [3], Ramona Munoz [1], Ghufran Ahmad [1], Ashish Verma [1], Cesar Acosta [4], Karolina Kaczmarski [1], Nicholas DePace, Jr. [1], Michael E. Goldis [5] and Joe Colombo [1,2,6,*]

1   Franklin Cardiovascular, Autonomic Dysfunction and POTS Center, Sicklerville, NJ 08081, USA; nicholasdepace@aol.com (N.L.D.); rmunoz@franklincardio.com (R.M.); ghufran.kmc@gmail.com (G.A.); ashish@ashishverma.net (A.V.); kikaczmarski@gmail.com (K.K.); nicholas.depace@gmail.com (N.D.J.)
2   NeuroCardiology Research Center, Sicklerville, NJ 08081, USA
3   New Jersey Heart, Sicklerville, NJ 08081, USA; drlou214@icloud.com
4   Wyckoff Heights Medical Hospital, Brooklyn, NY 11237, USA; drcesjr@gmail.com
5   Primary Care and Geriatrics, Stratford, NJ 08084, USA; goldisgeriatrics@yahoo.com
6   Physio PS, Inc., Atlanta, GA 30339, USA
*   Correspondence: jcolombo@physiops.com

**Abstract:** The progression of autonomic dysfunction from peripheral autonomic neuropathy (PAN) to cardiovascular autonomic neuropathy, including diabetic autonomic neuropathy and advanced autonomic dysfunction, increases morbidity and mortality risks. PAN is the earliest stage of autonomic neuropathy. It typically involves small fiber disorder and often is an early component. Small fiber disorder (SFD) is an inflammation of the C-nerve fibers. Currently, the most universally utilized diagnostic test for SFD as an indicator of PAN is galvanic skin response (GSR), as it is less invasive than skin biopsy. It is important to correlate a patient's symptoms with several autonomic diagnostic tests so as not to treat patients with normal findings unnecessarily. At a large suburban northeastern United States (Sicklerville, NJ) autonomic clinic, 340 consecutive patients were tested with parasympathetic and sympathetic (P&S) monitoring (P&S Monitor 4.0; Physio PS, Inc., Atlanta, GA, USA) with cardiorespiratory analyses, and TMFlow (Omron Corp., Hoffman Estates, Chicago, IL, USA) with LD Technology sudomotor test (SweatC™). This is a prospective, nonrandomized, observational, population study. All patients were less than 60 y/o and were consecutively tested, analyzed and followed from February 2018 through May 2020. P&S Monitoring is based on cardiorespiratory analyses and SweatC™ sudomotor testing is based on GSR. Overall, regardless of the stage of autonomic neuropathy, SweatC™ and P&S Monitoring are in concordance for 306/340 (90.0%) of patients from this cohort. The result is an 89.4% negative predictive value of any P&S disorder if the sudomotor GSR test is negative and a positive predictive value of 90.4% if the sudomotor testing is positive. In detecting early stages of autonomic neuropathy, P&S Monitoring was equivalent to sudomotor testing with high sensitivity and specificity and high negative and positive predictive values. Therefore, either testing modality may be used to risk stratify patients with suspected autonomic dysfunction, including the earliest stages of PAN and SFD. Moreover, when these testing modalities were normal, their high negative predictive values aid in excluding an underlying autonomic nervous system dysfunction.

**Keywords:** peripheral autonomic neuropathy; advanced autonomic dysfunction; diabetic autonomic neuropathy; cardiovascular autonomic neuropathy; small fiber disorder; sudomotor testing

## 1. Introduction

The progression of autonomic dysfunction from peripheral autonomic neuropathy (PAN) to cardiovascular autonomic neuropathy, including diabetic autonomic neuropathy and advanced autonomic dysfunction, increases morbidity and mortality risks [1,2]. Due to disagreements as to whether autonomic neuropathy is a function of the aging process

and its progression in patients under the age of 65, this study includes only patients under the age of 60 to omit the geriatric population. The increase in morbidity and mortality risks associated with the geriatric population and underlying autonomic dysfunction increases the risks of cardiovascular disease and renal disease as well as various multi-organ system disorders that contribute to numerous symptoms [3–15], thereby increasing medication-load, hospitalizations and healthcare costs. Earlier detection of autonomic dysfunction enables earlier treatment and a higher likelihood of slowing or staying the progression of autonomic dysfunction [16]. Autonomic neuropathy often includes orthostatic dysfunction [1,2] that affects cardiac and cerebral perfusion, leading to clinical symptoms including lightheadedness, dizziness, brain fog, cognitive and memory difficulties, sleep dysfunction, tension and migraine headache disorders and cranial sensory dysfunction. These symptoms alert clinicians to the need to test for autonomic dysfunction.

Autonomic dysfunction increases with age [16] and is accelerated by chronic disease and trauma, regardless of whether the trauma is psychologic or physiologic [16]. The accepted stages of autonomic neuropathy are—in order of severity: PAN, diabetic autonomic neuropathy and cardiovascular autonomic neuropathy [1,2]. In a standard autonomic function test, deep breathing, Valsalva, and head-up postural change (tilt-test or standing) are the challenges used to determine autonomic function as compared with the resting baseline. The earliest stage of autonomic dysfunction (PAN) is indicated by either or both deep breathing (see Figure 1) or Valsalva (see Figure 2) abnormalities. Diabetic autonomic neuropathy is indicated when either or both of the resting autonomic (parasympathetic or sympathetic) responses fall below normal, but the parasympathetic response is still >0.1 bpm$^2$ (see Figure 3) [16]. Diabetic autonomic neuropathy has been labeled advanced autonomic dysfunction for the same stage of autonomic neuropathy in patients not diagnosed with diabetes (see Figure 3) [16]. Cardiovascular autonomic neuropathy is indicated when the resting parasympathetic measure is extremely low (<0.1 bpm$^2$, see Figure 3) [16]. If autonomic dysfunction is detected early and treated [17], its progression may be slowed or stayed, regardless of the stage of autonomic neuropathy.

As PAN is the early stage of autonomic neuropathy, it typically involves small fiber disorder (SFD), often as an early component. SFD is an inflammation or deficiency of the C-nerve fibers which carry sympathetic and pain signals to and from the periphery. The inflammatory state of SFD is typically the early stage and the deficiency state of SFD is the later stage. The sympathetic nerve fibers affect peripheral vasoconstriction and sweat gland function, thereby affecting temperature control and wound healing. Currently, the preferred test for SFD as an indicator of PAN is galvanic skin response (GSR), as it is much more readily available, and far less invasive than a skin biopsy. GSR measures are used to indicate small fiber function [19–22]. GSR succeeded earlier tests, including Q-SART, Q-Sweat and thermoregulatory sweat testing as it is less time- and technician-intensive. There are multiple galvanic skin conduction testing modalities to assess sweat gland function [23–29]. SweatC™ is a GSR device and does not use, nor is it, an electrochemical skin response device. It captures quantitative sudomotor responses to assess the integrity of the post-ganglionic sudomotor nerves along the axon reflex. SweatC™ was used exclusively in this study as a type of quantitative sudomotor axon reflex test. While GSR has become a standard, there are still concerns that non-invasive sweat gland function may fail to demonstrate a conclusive measure of small fiber function [30,31]. Two technologies, Quantitative Sudomotor Axon Reflex Test (QSART) and Sympathetic Skin Response (SSR), both have overwhelming amounts of data supporting their use as medically necessary for the evaluation of autonomic dysfunction [32]. This study considers a complimentary and alternate approach to detecting PAN and thereby SFD or C-nerve Fiber function and compares it to sudomotor function as the current standard.

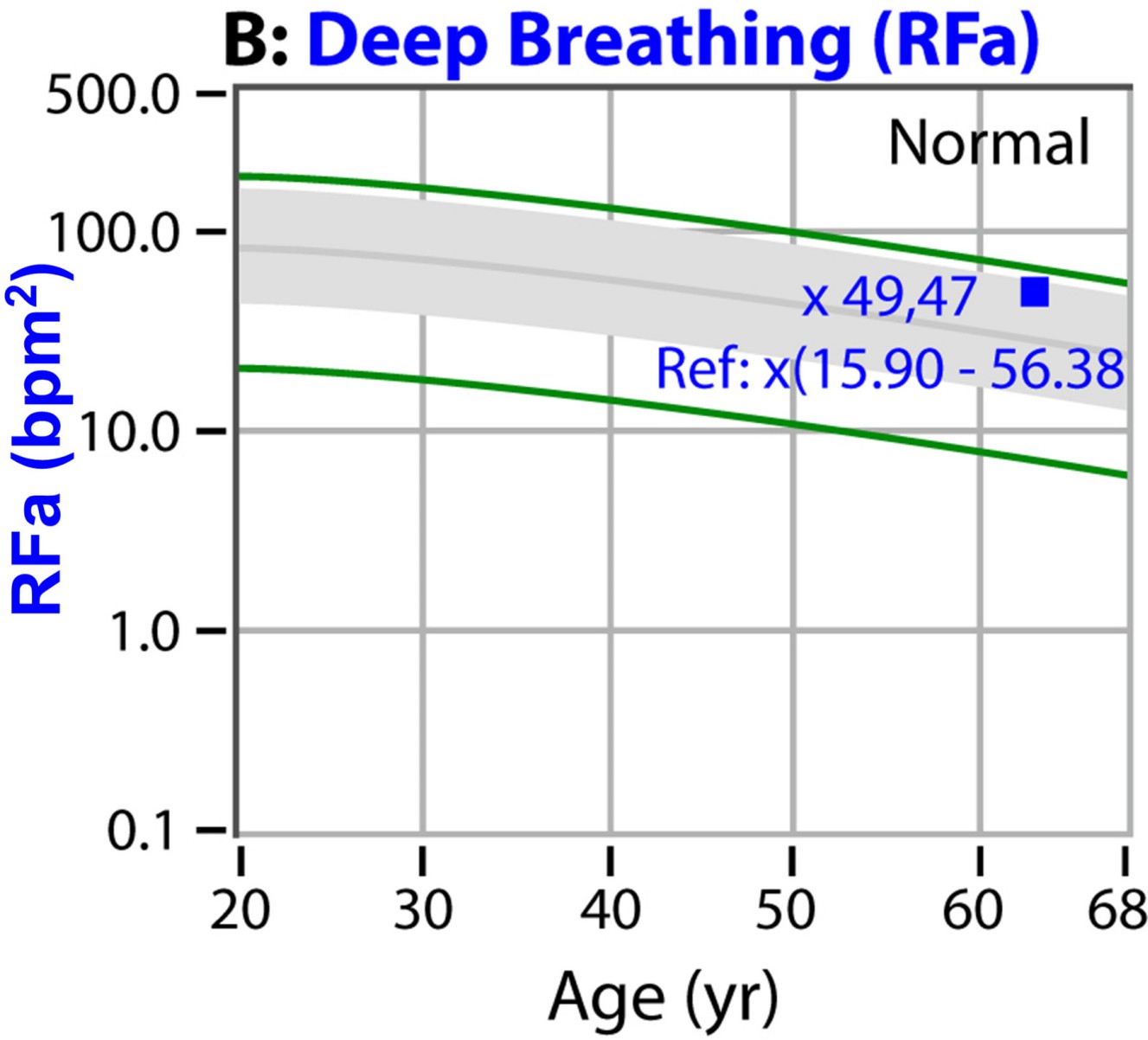

**Figure 1.** "Normal Response to Deep Breathing." The deep breathing challenge is a well-known parasympathetic stimulus, and arguably one of the strongest of parasympathetic stimuli. The figure displays a deep breathing response plot demonstrating a normal parasympathetic response for a 63 y/o patient. The gray area indicates the normal region [18]. These data are both age-and baseline-adjusted [16]. The 'x' preceding a number indicates that that number is baseline adjusted. The area between the green lines and the gray indicates borderline normal responses. Borderline low to low deep breathing responses are some of the earliest signs of PAN and risk of sudomotor abnormality or SFD.

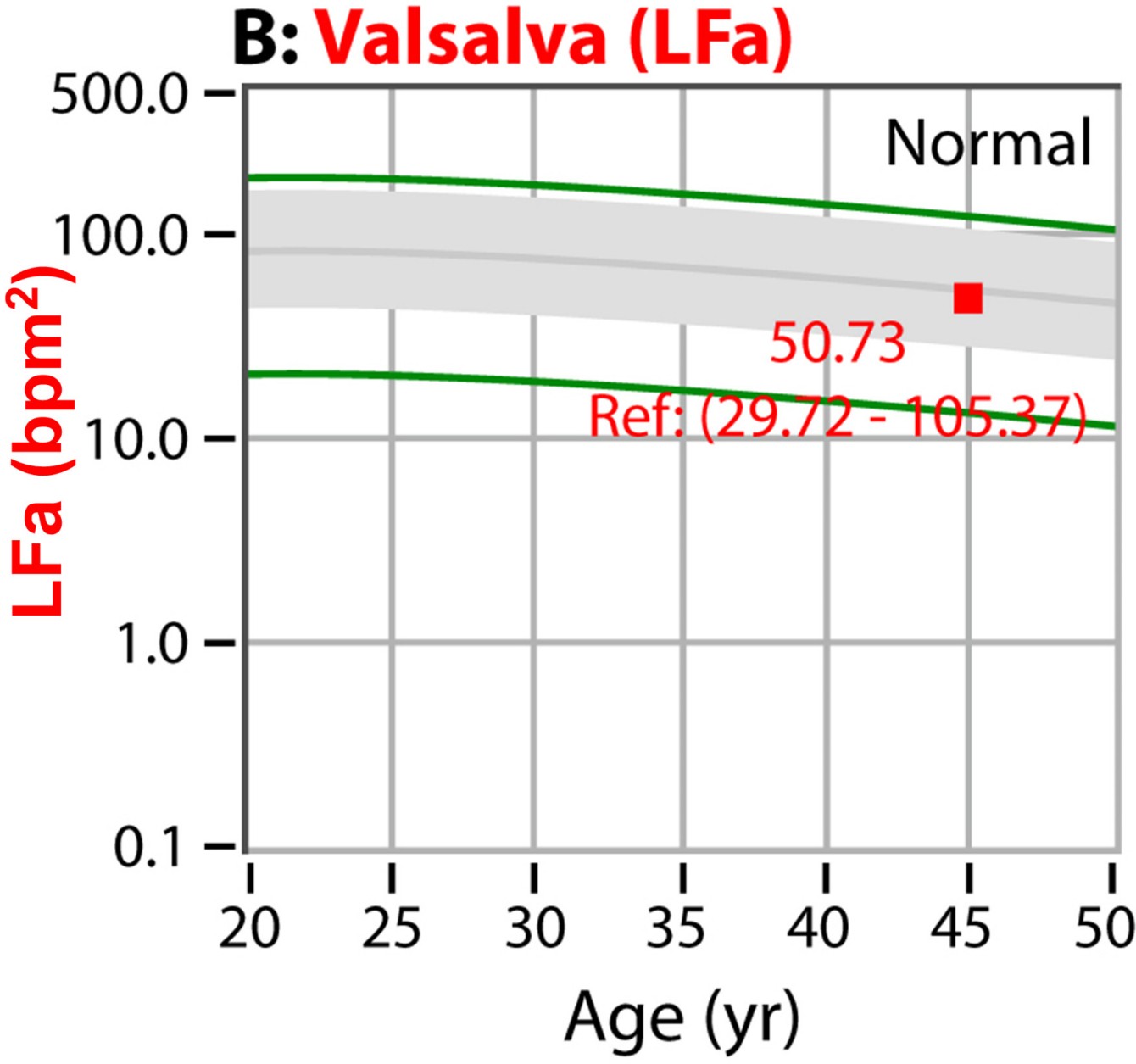

**Figure 2.** "Normal Response to Valsalva." The Valsalva challenge is a series of short (≤15 s) Valsalva maneuvers. Short Valsalva maneuvers are well-known and potent sympathetic stimuli. The figure displays a Valsalva response plot demonstrating a normal sympathetic response for a 45 y/o patient. The gray area indicates the normal region [18]. These data are both age- and baseline-adjusted [16]. The area between the green lines and the gray indicates borderline normal responses. Borderline low to low Valsalva responses are some of the earliest signs of PAN and risk of sudomotor or SFD.

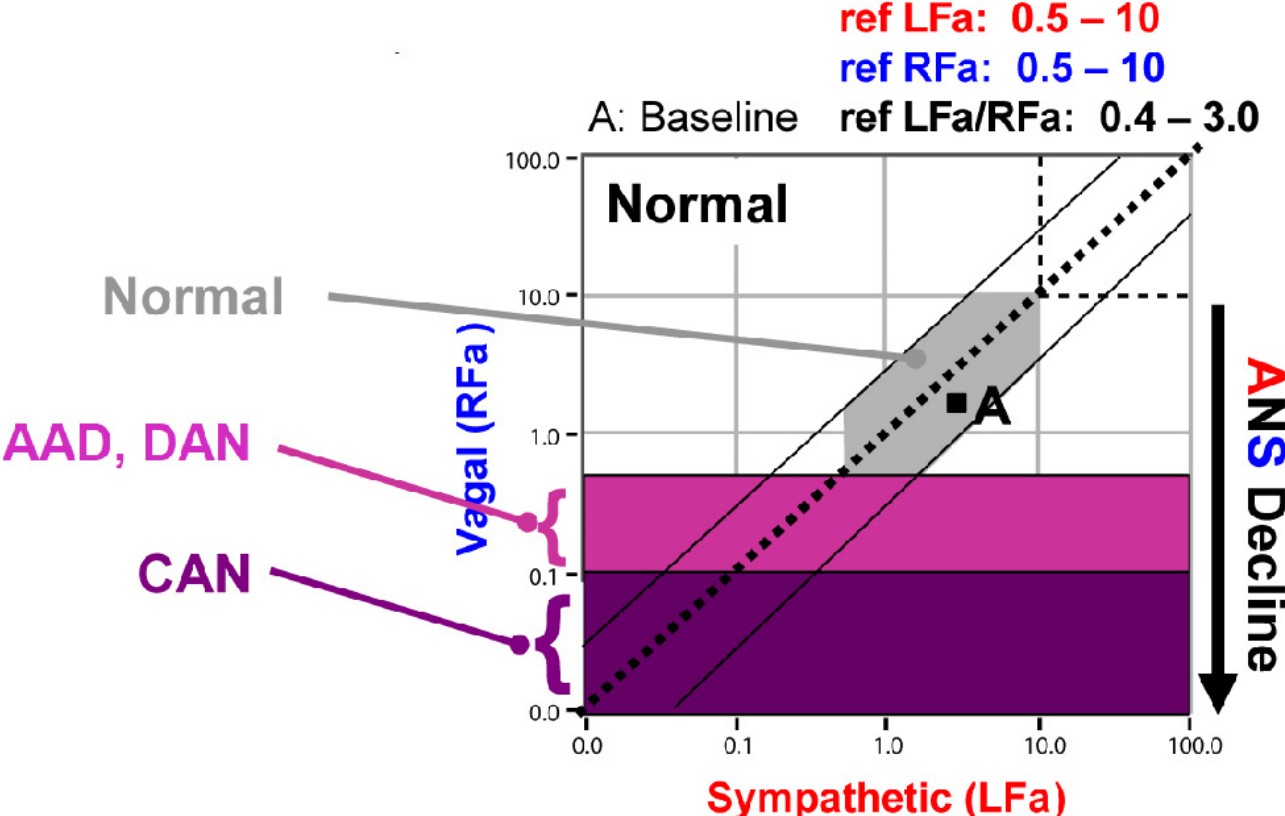

**Figure 3.** "Normal at Rest" An example resting (baseline) parasympathetic and sympathetic (P&S) response plot. The gray area depicts the normal response region. The purple highlighted areas depict the definitions of advanced autonomic dysfunction (AAD, light purple), diabetic autonomic neuropathy (DAN, also light purple) and cardiovascular autonomic neuropathy (CAN, dark purple). AAD and DAN indicate morbidity risk and CAN indicates mortality risk. Risk is stratified by sympathovagal balance. the space between the two outer diagonal lines defines normal sympathovagal balance ("LFa/RFa"). Regardless of resting autonomic state, normal sympathovagal balance normalizes morbidity and mortality risks. Above and to the left of the upper diagonal line indicates low sympathovagal balance which is a resting parasympathetic excess. Below and to the right of the lower diagonal line indicates high sympathovagal balance which is a resting sympathetic excess.

## 2. Methods

From a suburban cardiovascular and autonomic dysfunction practice in the northeastern United States (Sicklerville, NJ, USA), with patients drawn from around the world, 340 patients presenting with more than four autonomic symptoms [33] were consecutively tested and followed between February 2018 to September 2020. This cohort included 225 Females (66.0%), with an average age of 36.5 years (ranging from 14 to 59 years), and an average BMI of 26.7 #/in$^2$. All patients were tested with P&S Monitoring (P&S Monitor 4.0; Physio PS, Inc., Atlanta, GA, USA), which includes cardiorespiratory analysis, and TMFlow (Omron Corp., Hoffman Estates, Chicago, IL, USA), which includes the LD Technology Sudomotor SweatC™ test. This is a prospective, nonrandomized, observational, population study. PAN and SFD results from both test modalities are compared. All patients provided consent for their data to be included in this large population study and patient data were maintained according to HIPAA guidelines.

P&S Monitoring collects EKG, respiratory activity from chest electrodes and BP during four challenges: (1) rest (baseline, 5-min), (2) deep breathing (0.1 Hz, a parasympathetic chal-

lenge, 1-min), (3) short Valsalva maneuvers (<15 s, as a sympathetic challenge, 1:35-min), and (4) head-up postural change (stand, which is equivalent to tilt, 5-min [34]). With spectral analyses these data are analyzed, and independent and simultaneous P&S activity is measured throughout the clinical study [16]. Weakness in response to either or both deep breathing and Valsalva (collectively known as the breathing challenges) are the first signs of PAN, including SFD, perhaps even before overt symptoms of SFD. Normal and abnormal P&S response plots are depicted in Figures 1–4, respectively [16].

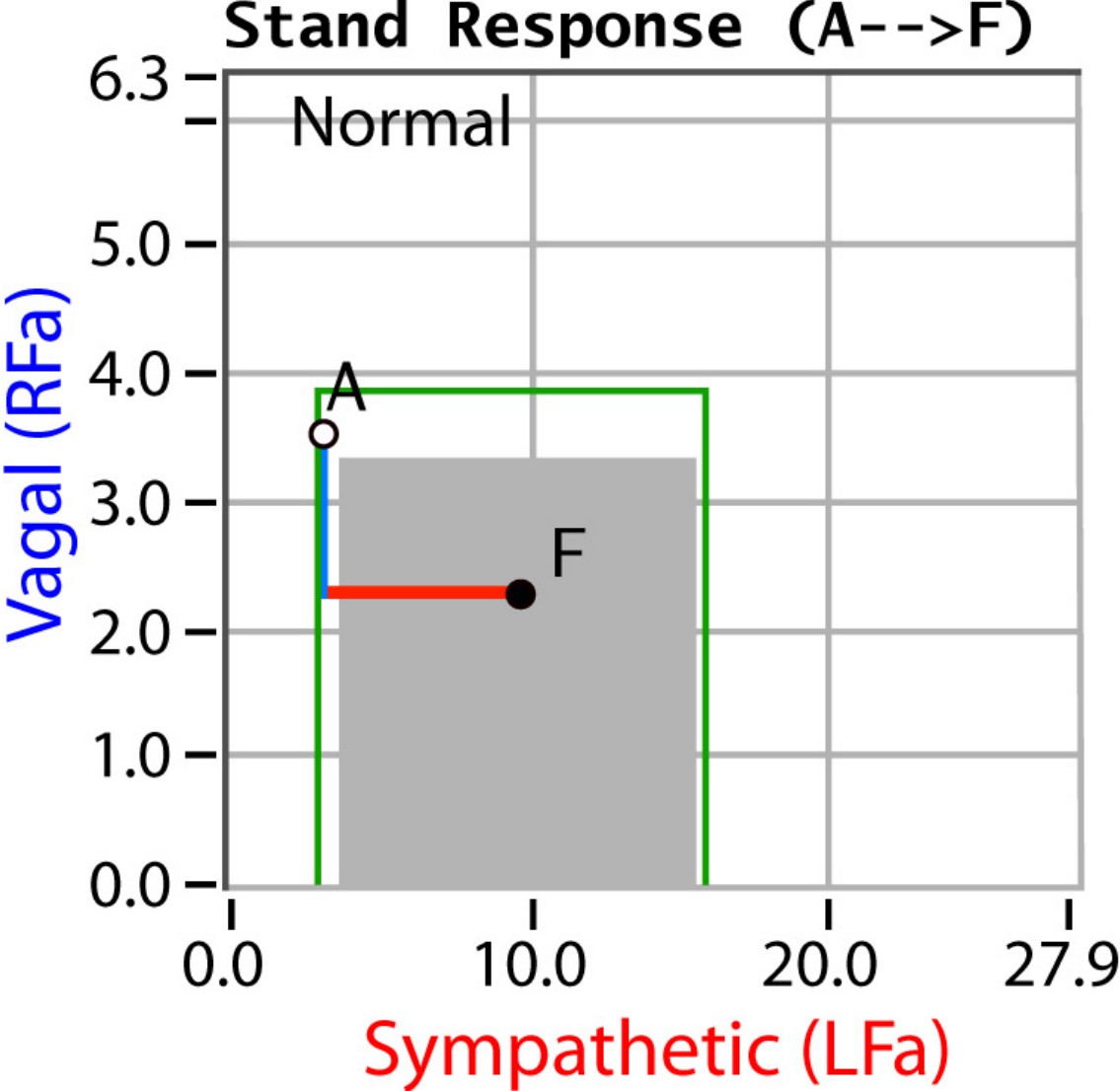

**Figure 4.** "Normal Response to Stand or Head-Up Postural Change." The stand challenge response provides information regarding causes of lightheadedness (e.g., orthostatic dysfunction as associated with and abnormally low sympathetic response, or syncope as associated with an abnormally high parasympathetic response), as well as an indication of the ability of the two autonomic branches to coordinate responses, not only to postural change but in the control and coordination of organs and organ systems [16]. The figure displays a stand response plot demonstrating a normal Parasympathetic (a decreased response from rest, 'A', to stand, 'F') with a normal Sympathetic response (an increased response from rest, 'A', to stand, 'F'). The gray area indicates the normal region. These data are baseline-adjusted. The area between the green lines and the gray indicates borderline normal responses. The symptoms associated with abnormal stand responses are often the first symptoms recognized as the results of autonomic dysfunction [16].

LD Technology SweatC™ test is not an electrochemical skin response. It does not use stainless steel electrodes for the hands and feet, nor does it use incremental voltages. Instead, it uses constant voltage and cloth (textile) electrodes placed on the feet, thus no electrochemical reactions may be obtained, and therefore it cannot induce any reverse ionophoresis. Sudomotor testing is based on electrochemical skin conductance analysis of the function of sweat glands from the bottom of the feet measured for 2 min [31]. Low sudomotor sweat peaks indicate decreased density of active cholinergic nerve fibers. Elevated sudomotor sweat peaks indicate inflammation.

The autonomic testing environment was a quiet, out of the way examination space, maintained at 70 °F and patients were permitted up to an hour to rest and acclimate. Technicians took steps to help patients remain calm and feel secure throughout the tests, including diming the lights for those who are light-sensitive. Pearson correlation and Student's *t*-test statistics are based on SPSS v. 20.

## 3. Results

The SweatC™ sudomotor test and the P&S Monitor indication of PAN both are indications for SFD. Table 1 provides a breakdown of the results from the two tests. From Sudomotor testing, 144 patients demonstrated abnormal results, indicating SFD from either inflammation or from depletion. Of the 144 patients demonstrating abnormal SweatC™, 82 demonstrated inflammation (high SweatC™ responses), and 67 demonstrated depletion (low SweatC™ responses); five patients demonstrated high and low SweatC™ responses, one foot each. The remaining 196 patients demonstrated normal SweatC™ responses. From P&S Monitoring, 122 patients demonstrated PAN (low breathing challenge responses), and 86 patients demonstrated advanced autonomic dysfunction, including PAN, totaling 208 abnormal indications from P&S Monitoring. The remaining 132 patients did not demonstrate autonomic neuropathy. Pearson's Correlation and Student's *t*-test indicate a statistically significant similarity between the results of the two tests (see Table 1).

**Table 1.** Sudomotor and P&S responses.

| Total Pop. = 340 | Inflammation/PAN | Depletion/AN | Total Abn | Total Nml |
|---|---|---|---|---|
| Sudomotor | 82 | 67 | 144 * | 196 |
| P&S | 122 | 86 | 208 | 132 |
| r | 0.960 | 0.894 | 0.802 | 0.802 |
| p | 0.051 | <0.001 | <0.001 | <0.001 |

* Five (5) patients demonstrated both inflammation and depletion.

Table 2 compares the two technologies, considering only the specific indication for PAN (abnormal breathing challenge responses). Both testing modalities indicate the same 130 (38.3%) patients as SFD positive, and 118 (34.7%) patients negative for SFD. There are only 14 (4.1%) patients for which SweatC™ did not indicate SFD and P&S Monitoring did indicate SFD (classified as false negative). There are 78 (34.7%) patients for which SweatC™ did indicate SFD and P&S Monitoring did not indicate SFD (classified as false positive). In summary, when considering only the specific indication for PAN, both technologies have a high (89.4%) negative predictive value and a low positive predictive value (62.5%).

**Table 2.** Comparison of SFD indications from SweatC™ sudomotor testing (sudomotor) and P&S monitoring (P&S), for the entire cohort, based on the specific definition of PAN. See text for details.

| *n* = 340 | P&S Positive | P&S Negative |
|---|---|---|
| Sudomotor Positive | 130 (38.3%) | 78 (22.9%) |
| Sudomotor Negative | 14 (4.1%) | 118 (34.7%) |

Upon further investigation, 58 (17.1% of the cohort) of the 78 patients for which SweatC™ did indicate SFD and P&S Monitoring did not (false positives), P&S Monitoring also documented advanced autonomic dysfunction, diabetic autonomic neuropathy or cardiovascular autonomic neuropathy. These were the only patients within the cohort to demonstrate these advanced stages of autonomic neuropathy. These advanced stages of autonomic neuropathy inherently include PAN as an earlier phase. However, the P&S Monitor computation of the deep breathing challenge and Valsalva challenge responses are baseline adjusted [16] for clinical accuracy. This adjustment may artificially normalize the challenge responses, relatively speaking. Removing the baseline adjustment to demonstrate that PAN is an inherent part of the more advanced stages of autonomic neuropathy, reveals that all 58 of those patients do indeed demonstrate PAN. Table 3 reanalyzes the results of the cohort with these 58 patients reclassified as SFD. As a result, the percent positive increases to 55.3%, and the percent false positive drops to 5.9% (the false negative and negative percentages do not change; therefore, the negative predictive value remains 89.4%) but the positive predictive value is now 90.4%.

**Table 3.** Comparison of SFD indications from SweatC™ sudomotor testing (sudomotor) and P&S Monitoring (P&S) considering that the more advanced autonomic neuropathies also involved PAN. In other words, reclassifying patients demonstrating advanced autonomic dysfunction, diabetic autonomic neuropathy or cardiovascular autonomic neuropathy as also demonstrating peripheral autonomic neuropathy and therefore small fiber disorder (P&S Positive). See text for details.

| *n* = 340 | P&S Positive | P&S Negative |
|:---:|:---:|:---:|
| Sudomotor Positive | 188 (55.3%) | 20 (5.9%) |
| Sudomotor Negative | 14 (4.1%) | 118 (34.7%) |

With this correction there is a high concordance rate (306/340 or 90.0%) and association between SweatC™ sudomotor positive indications and P&S Monitoring for all autonomic dysfunctions. In other words, abnormal physiology of small fibers is assessed by all types of P&S malfunction 90% of the time. Of the remaining 34 patients, 20 are classified as false positive, where SweatC™ indicates SFD and P&S Monitoring does not, and there continue to be 14 patients classified as false negative. The results from the re-analysis are: (1) a specificity of 85.5% (118/138 patients), (2) a sensitivity of 93.1% (188/202 patients), (3) a positive predictive value of 90.4% (188/208 patients) and (4) a negative predictive value of 89.4% (118/132 patients).

## 4. Discussion

In this study it is important to note that P&S Monitoring does not differentiate small fiber inflammation from deficiency. Both forms of SFD are included in the analyses since both are involved in SFD. In addition, for the purposes of this study, GSR is considered a very accurate test of SFD and is used as the gold standard even though there is some controversy. Sudomotor testing, as from SweatC™, measures the GSR at the positive electrode and the sweat peaks estimate of cholinergic nerve fiber density. Sweat peaks are calculated from the peak amplitude of the GSR at the positive electrode of patients resulting from sweating. Although sweat glands are controlled by the sympathetics, sweating may be influenced by other factors, including daily hydration (or dehydration). While many autonomic patients drink plenty of water a day they remain largely dehydrated because the water does not stay in their vasculature. Furthermore, many drinks intended to hydrate (i.e., sports drinks), actually dehydrate due to sugar, sugar substitutes, caffeine and alcohol (especially if one is not running around burning the sugar). In addition, many patients diagnosed with hypertension are also prescribed diuretics, when the hypertension may be

compensatory for orthostatic dysfunction. Significant dehydration may lead to anhidrosis and false positives, as defined above.

P&S Monitoring differs from all other autonomic monitors in that it is uniquely capable of measuring the two individual autonomic branches independently and simultaneously without assumption and approximation [35–38]. P&S monitoring permits follow-up testing and includes indications for PAN (including small C-fiber disorder) as well as P&S dysfunctions (including autonomic neuropathies) not detected by typical autonomic monitors. Such P&S dysfunctions include: (1) sympathetic withdrawal (an alpha-adrenergic insufficiency upon assuming a head-up posture, associated with orthostatic dysfunction; see Figure 4) [39] and (2) parasympathetic excess (an excessive cholinergic response to a stress, as modeled by Valsalva challenge or upon assuming a head-up posture, associated with parasympathetic or vagal over-reactions) [40]. Both of these autonomic dysfunctions may result in poor cerebral perfusion and symptoms of lightheadedness, some of the first symptoms reported that are associated with autonomic dysfunction.

The percent of patients classified as false negative (P&S monitoring indicating SFD and sudomotor testing not indicating SFD), while very low, may highlight the fact that P&S monitoring is typically a leading indicator of autonomic dysfunction and neuropathy, involving abnormal sudomotor results. Abnormal sudomotor testing must wait until SFD is present, and further, wait until it is significant enough before detecting significantly abnormal changes in skin conductance. P&S monitoring may detect these changes in the autonomic nervous system earlier, before resulting symptoms are significant. This may help to treat earlier and prevent symptoms and possibly slow the progression of autonomic dysfunction.

The percent of patients classified as false positive (P&S monitoring not indicating SFD and positive sudomotor testing indicating SFD) while also low, may highlight the fact that there are multiple reasons for sweating disorders, not all related to small fibers, such as connective tissue disorders, and medications, including some anti-depressants, antipsychotics, antihypertensives and opioids, all of which are common in today's culture.

Hypothetically, inflammation is an earlier stage of SFD and deficiency represents a more advanced stage of SFD. Typically, inflammation precedes deficiency in cells. The same is presumed here. Other nerve fibers may also be involved, including A-beta and A-delta nerve fibers. A-beta fibers carry touch information and feedback to the autonomic nervous system to signal incoming sensory information. A-delta fibers carry pain and temperature information and are known to be affected by blood flow [41]. The larger, more myelinated A-fibers are faster to respond than C-fibers and typically signal the acute ("sharp, specific") pain information and the C-fibers carry the chronic ("dull, diffuse") pain information.

Further studies are required to determine (1) if inflammation is associated with early stages of SFD or autonomic neuropathy (i.e., PAN, potentially more treatable if detected early); and (2) if deficiency is associated with later stages of SFD or autonomic neuropathy (i.e., advanced autonomic neuropathy or cardiovascular autonomic neuropathy, and may not be readily treatable). We believe that early detection may provide an advantage for reversal of SFD as well as autonomic neuropathy. Although further prospective studies are indicated, either test may be used alternatively by itself, which would be a cost saving measure to assess for autonomic dysfunction and evaluate for the presence of underlying risk factors (e.g., diabetes mellitus). Further prospective studies are needed.

## 5. Conclusions

In detecting SFD as an early stage of Autonomic Neuropathy, including Diabetic Autonomic Neuropathy, P&S Monitoring is comparable to sudomotor testing with high sensitivity, specificity and high positive and negative predictive values. Therefore, either testing modality may be used to risk stratify patients with suspected autonomic dysfunction, including the earliest stage, PAN, involving SFD. Likewise, these testing modalities when normal, with their high negative predictive values, may help to exclude an existing

autonomic dysfunction. Further prospective studies are needed to assess if any one study is sufficient to objectively diagnose patients with symptoms of autonomic dysfunction.

**Author Contributions:** All authors contributed significantly to the data collection, data analysis and development of this manuscript as confirmed by J.C. and N.L.D., Conceptualization and methodology, J.C. and N.L.D.; software, J.C.; validation, formal analysis, and investigation, J.C., N.L.D. and L.S.; resources, and data curation and collection, R.M., G.A., A.V., C.A., K.K., N.D.J. and M.E.G.; writing—original draft preparation, J.C.; writing—review and editing, N.L.D. and L.S.; visualization, supervision, project administration, J.C. and N.L.D.; funding acquisition, N/A. All authors have read and agreed to the published version of the manuscript.

**Funding:** This research received no external funding.

**Institutional Review Board Statement:** All patients provided consent for their data to be included in this large population study and patient data were maintained according to HIPAA guidelines.

**Informed Consent Statement:** Patient consent was waived due to all patients attending these clinics are informed that their data may be used for large population clinical studies, unless the patient objects. None of the 340 patients included in this study objected.

**Data Availability Statement:** All data are HIPAA protected and may be made available upon request.

**Conflicts of Interest:** Colombo is founder, and part owner of Physio PS, Inc., the provider of P&S Monitoring. There is no conflict with NeuroCardiology Research Corp., as it does not own anything involved in this study.

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
