# Peer review of "Parasympathetic and Sympathetic Monitoring Identifies Earliest Signs of Autonomic Neuropathy"

_neurosci, doi:10.3390/neurosci3030030_

Round 1
Reviewer 1 Report
This is a review of the P&S Monitoring system, for diagnosis of autonomic dysfunction with negative and positive predictive values when compared to the sudomotor testing. This is helpful as there are no other modalities that subdivide the ANS into sympathetic and parasympthetic components.
Overall, the paper was well written, and clear, but there is a major issue with the referencing. Multiple references are presented as ERROR messages; that will need to be cleared up.
Minor comments
Line 57 "Autonomic dysfunction declines with age" Should be either "Autonomic function decreases with age" or "Autonomic dysfunction risk increases with age"
line 59 "accepted stages of AN are In order..." remove in order as these are independent, not a definitive progression, or change to "in order of SEVERITY"
line 103 "As PAN is the early stage of AN.... and often as an early component" I'm not sure what this sentences is saying... what is an early component of what?
line 111 Please fix formatting of references all the way through
line 217 "Does not differentiate sweat gland inflammation from deficiency" This is confusing- P&S monitoring identifies abnormal sweat gland response due to denervation, correct? So how does that relate to inflammation vs deficience.
219 "is used as the standard" should be "is used as the gold standard"
line 272. Do you have data that supports inflammation as an early stage of SFD and deficiency as a more advanced stage? Or is that just an assumption? Not clear that pathophysiology is known.
Author Response
Reviewer #1, on behalf of all of the authors, I thank you for your kind words and constructive critique regarding our manuscript.
- The 13 “Error” messages from MSWord are corrected. Most of them were Figure and Table references. The two manuscript references referred to reference 16 [xvi].
- Line 57: “declines” is modified to “increases.”
- Line 59: “in order” is modified to “in order of severity.”
- Line 104-5: “As PAN is the early stage of Autonomic Neuropathy, it typically involves Small Fiber Disorder (SFD) and often as an early component.” is modified to “PAN typically involves Small Fiber Disorder (SFD).”
- All references have been checked to ensure superscripting has been removed.
- Line 217-8: “differentiate sweat gland inflammation from deficiency” is modified to “differentiate small fiber inflammation from deficiency”
- Line 219: “gold” is added.
- Line 272: has been modified to clarify the assumptions upon which the hypothesis is based.
Reviewer 2 Report
Review: DePace et al., Parasympathetic and sympathetic monitoring identifies earliest signs of autonomic neuropathy.
This manuscript looks at a number of different “physiological challenges” and its effects on galvanic skin response (GSR), as an indicator of autonomic dysfunction. While this is an interesting paper, and GSR may be one test that can be used to test autonomic dysfunction, it is a test that is highly affected by mood (fear and anxiety) and the testing environment. The authors do not discuss whether any of this was controlled for in their study. The description of the subject pool needs additional details and a statement that the patients gave consent needs to be added. There are many details missing in the methods. The discussion also could be expanded to include information regarding A-delta fiber function which is affected by blood flow, and A beta fibers, which feedback to the autonomic nervous system to signal incoming sensory information.
Edits:
Abstract:
Lines 13 and 14: you probably don’t need to capitalize the names of the various neuropathies (unless requested by the journal)
Line 16 please change “as” to “is”
Lines 21-24, please make sure you spell out all acronyms
Is it sudomotor or pseudomotor testing?
Introduction
Lines 45-46: What is the evidence that people go from being non-geriatric to geriatric between ages 60 and 65?
The references are in roman numerals? Is this correct?
Line 57: Do the authors mean autonomic dysfunction increase with age, or autonomic function decline with age?
Lines 64, 67, 70 etc: why does it say error, reference source not found? Should another reference go there or is this a note that was not deleted when the reference was added?
Figures 1, 2….please add an axis title to the y-axis of these figures. Why are the age ranges different for these two graphs?
Figure 3, please write out what a “P&S” response is.
The authors need to go through the manuscript and determine when to capitalize words and when not to.
Line 106: do the authors mean the inflammatory “stage” of SFD?
Line 112: please change “conductions” to “conduction”
Methods
The methods need to describe the testing environment. Autonomic function is affected by temperature, noise and a number of other stimuli. Were patients allowed to rest and acclimate to the testing environment prior performing the test? Where were measurements collected from (where were sensors placed)? How long were measurements collected during each challenge?
Discussion:
Line 272: This first sentence makes no sense.
Author Response
Reviewer #2, on behalf of all of the authors, I thank you for your kind words and constructive critique regarding our manuscript.
- Line 13-4, and throughout: thank you and we’ll leave the capitalization of the neuropathies to the journal.
- Line 16: “as” is modified to “is”
- Lines 21-4: the abbreviation P&S is spelled out, the rest are names.
- It is sudomotor testing (no ‘p’)
- Line 45-6: the “geriatric” issue is addressed in the modification in Line 47.
- Roman numeral references are what we received back from the publisher. We assume it is their preference.
- Line 57: “autonomic dysfunction increases with age” as modified.
- The 13 “Error” messages from MSWord are corrected. Most of them were Figure and Table references. The two manuscript references referred to reference 16 [xvi].
- 1 & 2, y-axis labels are added and the age ranges are different because the data are from two different (random) subjects.
- 3: P&S is written out (Parasympathetic and Sympathetic)
- Capitalization has been addressed
- Line 106: we mean “state”
- Line 112: the ‘s’ is removed.
- RE: Methods comments – the last two paragraphs of the original Methods are modified to include time challenges and placement of sensors. An additional paragraph was added to describe the testing conditions.
- Line 272: has been modified to clarify the assumptions upon which the hypothesis is based.
- The discussion is expanded to include information regarding A-delta and A beta fiber function as other nerve fiber that are possibly involved in peripheral autonomic neuropathy.
Round 2
Reviewer 2 Report
The authors addressed the reviewers concerns. They added additional information to the methods and discussion and made editorial changes.